# Red Sea Bream Iridovirus (RSIV) Kinetics in Rock Bream (*Oplegnathus fasciatus*) at Various Fish-Rearing Seawater Temperatures

**DOI:** 10.3390/ani12151978

**Published:** 2022-08-04

**Authors:** Kyung-Ho Kim, Kwang-Min Choi, Min-Soo Joo, Gyoungsik Kang, Won-Sik Woo, Min-Young Sohn, Ha-Jeong Son, Mun-Gyeong Kwon, Jae-Ok Kim, Do-Hyung Kim, Chan-Il Park

**Affiliations:** 1Department of Marine Biology & Aquaculture, Institute of Marine Industry, College of Marine Science, Gyeongsang National University, 2, Tongyeonghaean-ro, Tongyeong 53064, Korea; 2Aquatic Disease Control Division, National Fishery Products Quality Management Service, 216, Gijanghaean-ro, Gijang, Busan 46083, Korea; 3Aquatic Disease Control Division, National Fishery Products Quality Management Service, 17, Jungnim 2-ro, Tongyeong 53019, Korea; 4Department of Aquatic Life Medicine, College of Fisheries Science, Pukyong National University, 45, Yongso-ro, Nam-Gu, Busan 48513, Korea

**Keywords:** *Oplegnathus fasciatus*, red sea bream iridovirus, seawater temperature, viral shedding, viral kinetics

## Abstract

**Simple Summary:**

Red sea bream iridoviral disease (RSIVD) generates serious economic losses by causing mass mortality events of rock bream during the season with high water temperature in the Republic of Korea and other Asian countries. However, very few studies have investigated RSIV kinetics in rock bream under various rearing water temperatures. In this paper, we investigated the viral load shedding of RSIV into seawater after artificially infecting rock bream (*Oplegnathus fasciatus*) with the virus. Overall, our data suggest that the viral load shedding of RSIV into seawater varies depending on water temperature and virus inoculation concentration. Our results reveal the potential of non-invasive virus detection approaches, such as the utilization of environmental DNA in fish farms. In addition, we showed that the quantitative analysis of seawater viruses can indirectly improve our understanding of disease progression in fish, potentially contributing to enhanced disease control.

**Abstract:**

Red sea bream iridoviral disease (RSIVD) causes serious economic losses in the aquaculture industry. In this paper, we evaluated RSIV kinetics in rock bream under various rearing water temperatures and different RSIV inoculation concentrations. High viral copy numbers (approximately 10^3.7^–10^6.7^ RSIV genome copies/L/g) were observed during the period of active fish mortality after RSIV infection at all concentrations in the tanks (25 °C and 20 °C). In the group injected with 10^4^ RSIV genome copies/fish, RSIV was not detected at 21–30 days post-infection (dpi) in the rearing seawater. In rock bream infected at 15 °C and subjected to increasing water temperature (1 °C/d until 25 °C) 3 days later, the virus replication rate and number of viral copies shed into the rearing seawater increased. With the decrease in temperature (1 °C/d) from 25 to 15 °C after the infection, the virus replicated rapidly and was released at high loads on the initial 3–5 dpi, whereas the number of viral copies in the fish and seawater decreased after 14 dpi. These results indicate that the number of viral copies shed into the rearing seawater varies depending on the RSIV infection level in rock bream.

## 1. Introduction

Red sea bream iridovirus (RSIV) was first detected in 1990 in red sea bream (*Pagrus major*) cultivated in Japan and caused a mass mortality event [1]. According to the International Committee on Taxonomy of Viruses (ICTV), RSIV has been identified as a new genus of *Megalocytivirus* within the family *Iridoviridae* [2], along with infectious spleen and kidney necrosis virus (ISKNV) [3,4], turbot reddish body iridovirus (TRBIV), and scale drop disease virus (SDDV) [5,6,7]. It has a double-stranded DNA genome approximately 110 kb in length, with an icosahedral virion capsid 140–200 nm in diameter [8].

This *Megalocytivirus* infection is widely distributed in Asian countries, causing serious economic damage to the aquaculture of various marine fish species [8]. Red sea bream iridoviral disease (RSIVD), caused by RSIV and ISKNV genotypes, is a disease managed by the World Organization for Animal Health (OIE) because it is highly pathogenic and affects more than 30 fish species [9]. Japanese red sea bream farms have been afflicted by outbreaks of RSIVD every summer since the virus was first reported [1]. In 1998, the virus was detected in rock bream (*Oplegnathus fasciatus*) cultured in the Republic of Korea during high summer temperatures [10,11]. A formalin-inactivated vaccine was developed in response to the RSIVD epidemic; however, it is not used in highly sensitive fish species such as rock bream because of its low disease prevention effect [9]. Recently, the RSIV concentration method using iron flocculation [12] has been developed. It is simpler and more economical than the concentration methods using ultrafiltration [13,14], ultracentrifugation [15], polyethylene glycol precipitation [16], and cationic coating filter methods previously employed for fish pathogenic viruses [17,18]. Horizontal transmission through seawater is considered to be the main route of transmission of RSIVD [9]; it is known that RSIV infectivity in environmental seawater at 15 °C can be maintained for 7 d [19]. This transmission hypothesis can be used for RSIV monitoring using virus concentrations in the environmental seawater of fish farms.

Rearing water temperature influences the incidence and severity of fish viral infections by directly altering viral replication and indirectly enhancing the efficacy of the host immune response [20,21]. Previous studies have demonstrated a clear correlation between mortality, water temperature, and RSIV infection in rock bream [22,23]. RSIV is a lethal viral disease in rock bream aquaculture in the Republic of Korea and other Asian countries and causes 100% cumulative mortality at high water temperatures [10]; however, there is little information on RSIV kinetics, and a quantitative analysis of the shedding of virus into rearing seawater.

In the present study, rock bream individuals were artificially infected with RSIV under various rearing water temperature conditions, and the virus replication pattern in rock bream and RSIV shedding in rearing seawater were monitored. In addition, changes in the RSIV replication pattern according to water temperature shifts were investigated by artificially increasing or decreasing the water temperature 3 days post infection (dpi) in rock bream maintained at 15 °C and 25 °C. Our findings will be useful for non-invasive detection of annual RSIV outbreaks in rock bream farms and help to establish measures to control the disease.

## 2. Materials and Methods

### 2.1. Experimental Fish

Rock bream individuals were obtained from a hatchery in Namhae (Namhae, Gyeongsangnam-do, Korea) and reared at Gyeongsang National University. Fifteen fish were randomly selected to confirm that the fish from the hatchery was free of bacterial, parasitic, and viral diseases. After collecting fish spleens and smearing them on the brain heart infusion agar medium, the presence of pathogenic bacteria was checked, and the gills and body surface were examined microscopically for the presence of parasites. The presence of RSIV was confirmed by real-time PCR of the DNA extracted from fish spleens. Approximately 1000 rock bream individuals (mean length ± SD: 10.4 ± 1.3 cm, mean weight ± SD: 26.8 ± 3.8 g) were acclimatized for 2 weeks in a 1600 L tank. The tank was a flow-through aquaculture system (500–1000 L/h), and the seawater was continuously supplied with sand-filtered, 50 μm filter-housed, and UV-treated (>30 mW/cm^2^) seawater. During the acclimatization period, seawater was maintained at a temperature of 21 ± 1 °C, dissolved oxygen at >6 mg/L, salinity at 28–30 psu, pH at 7.8–8.6, and NH_3_^+^ at <0.1 mg/L. In addition, rock bream individuals were fed a pellet/extruded type commercial diet (Suhyupfeed Co., Ltd., Uiryeong, Gyeongsangnam-do, Korea; 52% crude protein and 12% lipids) in the amount of 2–3% of the fish body weight per day.

### 2.2. Virus

In August 2019, the virus was obtained from the spleen and kidney of RSIV-infected rock bream, and the tissue samples were stored at −80 °C. The presence of RSIV was confirmed by PCR and sequencing analysis methods provided by the OIE [9]. RSIV was classified as RSIV genotype II (accession number: AY532608) by a phylogenetic analysis of the *MCP* gene sequence [24]. For the recovery of viral pathogenicity before the infection experiments [25], RSIV-infected tissues were homogenized with phosphate-buffered saline (PBS) and centrifuged at 3000× *g* for 20 min at 4 °C. After the virus-containing supernatant was filtered through a 0.45 μm syringe filter, it was intraperitoneally (IP) injected into the rock bream individuals kept in a tank at 25 °C.

### 2.3. Determination of RSIV Genome Copy Number

For RSIV genome copy number analysis, the spleen and kidney (25–50 mg) of rock bream individuals were removed, and genomic DNA (gDNA) was extracted using the AccuPrep^®^ Genomic DNA Extraction Kit (Bioneer, Daejeon, Korea) according to the manufacturer’s instructions. The gDNA extracted from rock bream and rearing seawater were analyzed by TaqMan probe-based real-time PCR targeting the *MCP* gene of RSIV [26]. The primer/probe set for RSIV detection consisted of RSIV RT F (5′-TGA CCA GCG AGT TCC TTG ACT T-3′), RSIV RT R (5′-CAT AGT CTG ACC GTT GGT GAT ACC-3′), and RSIV probe (5′-FAM-AAC GCC TGC ATG ATG CCT GGC-BHQ1-3′). Real-time PCR was performed in a final volume of 25 μL of reagent mixture containing 5 μL gDNA (50–100 ng/μL), 900 nM of each primer, 250 nM of probe, 15 μL of AccuPower Plus DualStar qPCR Master Mix (with UDG) (Bioneer), and 2 μL of nuclease-free water using a Thermal Cycler Dice^®^ Real Time System III (Takara, Kusatsu, Japan). The amplification thermal profile consisted of one cycle at 95 °C for 10 min (pre-denaturation), followed by 45 cycles at 95 °C for 15 sec (denaturation) and 60 °C for 1 min (annealing and extension). For the production of a positive control plasmid, gDNA was extracted from RSIV using the AccuPrep^®^ genomic DNA Extraction Kit (Bioneer) according to the manufacturer’s protocol. To prepare a recombinant plasmid as a positive control and generate real-time PCR standard curves, the 1362 bp PCR amplification product was ligated to a pGEM-T easy vector (Promega, Madison, USA) and added to plasmid-transformed *Escherichia coli* strain JM109 [24]. Thereafter, the plasmid containing the *MCP* gene amplification product in *E. coli* was purified using the GeneAll^®^ Exprep^™^ Plasmid SV Kit (GeneAll Biotechnology, Seoul, Korea). The copy number of purified plasmid DNA was calculated according to the formula described in a previous study [27].
$$Number\;of\;copies/\mu L=\;\frac{6.022\times 10^{23}\;\left(molecules/mole\right)\times DNA\;concentrations\;\left(g/\mu L\right)}{Number\;of\;base\;pairs\times 660\;daltons}$$

### 2.4. Iron Flocculation Method for RSIV Detection in Seawater

To measure RSIV recovery using the iron flocculation method, nine 10-fold serial dilutions of RSIV-spiked artificial seawater, ranging from 7.6 × 10^8^ to 7.6 RSIV genome copies/L, were prepared. Three replicates, each containing 500 mL of RSIV-spiked artificial seawater, were prepared for each concentration. Then, 50 μL of an FeCl_3_ solution (4.83 g FeCl_3_∙6H_2_O in 100 mL of distilled water) were added to the RSIV-spiked seawater, and the mixture was gently stirred using a magnetic stirrer for 2 h [12]. The flocculated virus was filtered through a 0.8 μm pore size polycarbonate filter (Whatman, Maidstone, UK) held on a filter holder with a receiver (Nalgene, New York, USA) under reduced pressure. The viruses collected in the polycarbonate filter were transferred to a 2 mL tube and stored at −80 °C until gDNA extraction. The filtered virus flocculate was directly extracted from the filter using the AccuPrep^®^ Genomic DNA Extraction Kit (Bioneer). In detail, 400 μL of the TL buffer provided with the gDNA extraction kit, 60 μL of proteinase K, and 20 μL of RNase A were added to the flocculate, and the mixture was incubated at 60 °C for 3 h. After 400 μL of GB buffer were added to the mixture, the mixture was briefly vortexed, incubated at 60 °C for 30 min, and diluted with 800 μL of absolute ethyl alcohol. Thereafter, DNA purification was performed according to the instructions of the Genomic DNA Extraction Kit (Bioneer), and final DNA elution was performed in 50 μL volume.

### 2.5. Experimental Infection at Different Rearing Water Temperatures and RSIV Concentrations

Fish were maintained at 25 °C, 20 °C, and 15 °C for 2 weeks before virus infection to determine the cumulative mortality and virus kinetics after RSIV infection at various rearing water temperatures and virus concentrations, as well as to detect the viral load in seawater. Subsequently, 0.1 mL of the RSIV solution (10^8^, 10^6^, and 10^4^ RSIV genome copies/fish) were IP-injected into the fish kept at 25 °C, 20 °C, and 15 °C (30 fish/50 L, approximately 16 kg/m^3^). In the control group, 0.1 mL PBS were IP-injected into 30 fish. Then, 50% of the rearing seawater was replaced daily with sand-filtered, 1 μm filter-housed, and UV-treated (>30 mW/cm^2^) seawater every day, and a commercial diet was fed to the fish as described above. Mortality was observed for 30 d. Rock bream spleens, kidneys, and rearing seawater (500 mL) were collected at 1, 3, 5, 7, 10, 14, 21, and 30 d after RSIV inoculation (*n* = 3 in each group). Rearing seawater was collected from the tanks of the experimental group to measure the mortality. Dead fish were collected daily, and real-time PCR was used to confirm whether mortality was the consequence of the RSIV infection. Experiments for RSIV kinetics analysis were performed as described above (Section 2.3 and Section 2.4).

### 2.6. Rearing Water Temperature Shift after Experimental Viral Infection

Rock bream (30 fish) were maintained in 50 L seawater at 25 and 15 °C for 3 weeks before RSIV infection. The group exposed to shifting temperatures in the rearing tanks was IP-injected with 0.1 mL of the RSIV solution containing 10^6^ RSIV genome copies/fish. The control group was IP-injected with the same amount of PBS. In the shifting-up group (15 to 25 °C), the rearing water temperature increased (1 °C/d) from 3 dpi until it reached 25 °C at 12 dpi. In the shifting-down group (25 to 15 °C), the rearing water temperature decreased (1 °C/d) from 3 dpi until reaching 15 °C at 12 dpi. In each group, then, 50% of the rearing seawater was replaced daily with sand-filtered, 1 μm filter-housed, and UV-treated >30 mW/cm^2^) seawater, and a commercial diet was fed to the fish as described above. Mortality was observed for 30 d. Experiments for RSIV kinetics analysis were performed as described above (Section 2.3 and Section 2.4).

### 2.7. Statistical Analysis

Statistical analysis was performed using GraphPad Prism (version 9.4.0). An ordinary one-way ANOVA with Dunnett’s correction was performed when comparing multiple groups. Significant differences were compared to controls when the virus was first detected within each group. Statistical significance was reported as follows: * *p* < 0.05; ** *p* < 0.01; *** *p* < 0.001; **** *p* < 0.0001.

## 3. Results

### 3.1. RSIV Recovery Efficiency of the Iron Flocculation Method

The mean RSIV recovery efficiency of the iron flocculation method at all concentrations was approximately 83% (7.6 × 10^8^ to 7.6 RSIV genome copies/L seawater) (Appendix A), and the relationship between the spiked number and recovery number of RSIV was linear (R^2^ = 0.9994) (Appendix A). These results were consistent with previously reported results of the RSIV concentration in seawater, as determined by the iron flocculation method [12].

### 3.2. Influence of Different Rearing Water Temperatures and RSIV Concentration on RSIV Kinetics

In the group maintained at 25 °C, 100% cumulative mortality of rock bream individuals infected with 10^8^, 10^6^, and 10^4^ RSIV genome copies/fish occurred at 12, 14, and 18 dpi, respectively (Figure 1A). In the group injected with 10^8^ RSIV genome copies/fish and maintained at 20 °C, the cumulative mortality was 100% at 26 dpi. In the group injected with 10^6^ and 10^4^ RSIV genome copies/fish, it was 76.6% and 33.3%, respectively, at 30 dpi (Figure 1A). Mortality was not observed in the group maintained at 15 °C after injections with any of the three RSIV concentrations. In the 25 °C and 20 °C groups, a lower virus concentration resulted in delayed mortality (Figure 1A). In all the challenge groups (i.e., groups where fish were injected with the RSIV), the higher the rearing water temperature and RSIV injection concentration, the more rapid the RSIV replication in the rock bream (Figure 2). The group maintained at 15 °C showed a higher level of kinetics than 10^7^ RSIV genome copies/mg at 21 dpi when inoculated with 10^8^ RSIV genome copies/fish (Figure 2C); however, mortality was not observed until 30 dpi (Figure 1A). The initial detection time of RSIV in seawater was different depending on the infection concentration and water temperature (Figure 2). After the RSIV injection, the RSIV shedding ratio in seawater increased as the RSIV kinetics of rock bream increased in all groups (Figures 2 and 4A). During the period of active mortality after the RSIV IP injection, the maximum RSIV shedding ratio was 10^3.7^ to 10^6.7^ RSIV genome copies/L/g in 25 °C and 20 °C groups, respectively (Figure 2A,B,D,E,G,H). In addition, the RSIV kinetics decreased after 21 dpi in the group injected with 10^8^ RSIV genome copies/fish at 15 °C and decreased after 14 dpi in the group injected with 10^6^ and 10^4^ RSIV genome copies/fish at 15 °C (Figures 2C,F,I and 4A). In the group injected with 10^4^ RSIV genome copies/fish at 15 °C, low RSIV kinetics (<10^2.3^ RSIV genome copies/mg) were observed after 21 and 30 dpi, and RSIV was not detected in the rearing seawater (Figure 2I).

### 3.3. Influence of Rearing Water Temperature Shift on RSIV Kinetics

Rock bream mortality because of RSIV infection was highly dependent on the exposure time to high water temperature. In the shifting-up group, 100% cumulative mortality occurred at 21 dpi (Figure 1B). The 100% cumulative mortality of the shifting-up group was delayed by 7 d compared to that of the group exposed to a constant temperature of 25 °C after RSIV infection; however, in the shifting-up group, 100% mortality occurred within 10 d after the temperature reached 25 °C (Figure 1B). After RSIV infection up to 7 dpi (when the rearing water temperature was below 20 °C), the RSIV kinetics and RSIV shedding ratio into the rearing seawater were at the level of 10^4.2^ RSIV genome copies/mg and 10^0.6^ RSIV genome copies/L/g, respectively (Figure 3A). The RSIV kinetics and RSIV shedding ratio into the seawater increased rapidly from the point when the rearing water temperature was >20 °C (Figure 3A and Figure 4B). The highest seawater shedding ratio was 10^7.9^ RSIV genome copies/L/g at 21 dpi, which showed a significant difference compared to the number of viral copies in the seawater at 5 dpi (*p* < 0.05) (Figure 3A). In the shifting-down group, 20% cumulative mortality was observed at 30 dpi (Figure 1B). After RSIV infection, the RSIV replication rapidly increased in a similar pattern to that in the group exposed to 25 °C constant temperature until 7 dpi; however, this was followed by a delay in viral replication (Figure 3B and Figure 4B). The highest RSIV kinetics and RSIV shedding ratio into the seawater were recorded at 14 dpi, and they then continued to decrease until 30 dpi (Figure 3B and Figure 4B). In the control group, mortality related to the water temperature shift was not observed (Figure 1B).

## 4. Discussion

Horizontal transmission through the aquatic environment is considered to be the main route of RSIV infection [9,28,29]. In the Republic of Korea, as rock bream is cultured under a net pen enclosure, virus particles shed from RSIVD-infected rock bream can move through seawater, causing damage to nearby farmed fish. In the present study, we found that the viral load in fish and seawater changes as RSIVD progresses. Although several studies have focused on the viral epidemiology in fish bodies, very few laboratory studies have been conducted to monitor viruses in rearing seawater. In this paper, we successfully detected the RSIV genome in the rearing water of virus-infected rock breams using the iron flocculation method (10^1.86^–10^8.50^ RSIV genome copies/L seawater, quantitative analysis results without considering the weight of the fish that survived in the tank) (Figure 2 and Figure 3). RSIV was first detected at 3–10 dpi, and the timing differed depending on the rearing water temperature and concentration of RSIV. Previous research has shown that, in RSIV-infected Japanese amberjack (*Seriola quinqueradiata*), the virus was detected from 3 dpi (10^2.5^ RSIV genome copies/L seawater) [12]. In Atlantic salmon (*Salmo salar* L.), salmonid alphavirus (SAV) is known to begin to be excreted through mucus and feces immediately after infection and may persist for 3–4 weeks thereafter [30]. Although no quantitative viral load analysis in feces and mucus was performed in the present study, the virus was not detected in the seawater immediately after the IP challenge, but was detected in seawater 1–5 days after RSIV was detected in the fish body. Although it is not possible to precisely define when the virus sheds into seawater, when a viral load in RSIV-challenged fish is above a certain level, it seems reasonable to hypothesize that the virus is shedding into seawater.

Viral shedding is an indicator of viral infection and multiplication in a host. Previously, a positive correlation was observed between viremia and viral shedding [31]. Our results show that the higher the inoculation concentration or optimal replication water temperature of the virus, the more virus was released. Viral shedding peaks have been reported to correlate with the time of fish death (immediately or at the time of death) [12,32]. Our study also suggests that high viral load peaks detected in the seawater were also associated with mortality. On the other hand, RSIV genome (10^−0.46^–10^2.84^ RSIV genome copies/L/g) was detected in the rearing seawater, even though mortality was not observed in RSIV-infected rock bream at 15 °C. In vivo and in vitro analyses have shown that the optimal temperature range for RSIV replication is 20–25 °C [33,34]. It is known that rock bream has a relatively low sensitivity to RSIV when the water temperature is below 17 °C [34]. In the present study, no virus was detected in the rearing water at 21 and 30 dpi in the group inoculated with 10^4^ RSIV genome copies/fish at 15 °C. This is likely related to the low viral load of the fish, which means that the virus has not been shed or that the viral load in the seawater is below the detection limit of the virus detection method. However, it has been reported that the surviving rock bream still carry RSIV 100 days after RSIV infection [34]. This indicates that rock bream can carry viruses at low water temperatures for a long time, which can be a major cause of repeated mass mortality in the rock bream industry as a consequence of RSIV infection during May–July (approximately 15–25 °C), when seawater temperatures increase sharply in rock bream farms in the Republic of Korea. In order to explain this phenomenon, additional experiments are needed to confirm whether the disease is transmitted through seawater by cohabiting naïve (RSIV-free) fish at the limit of detection in seawater. In addition, monitoring the amount of virus released from rock bream long time after the end of RSIV infection is also necessary.

In the present study, because the challenge experiments were performed in a closed aquaculture system rather than a flow-through aquaculture system, we may not have fully reproduced the disease progression in the aquaculture field. However, in a previous study, even when a large amount of rearing seawater in the tank was flushed with fresh seawater after virus infection in fish, the time points for the viral load detection in the fish and the peak of viral load in the seawater were consistent [35]. Although we could not fully reproduce the aquaculture environment in this study, we succeeded in detecting RSIV in seawater at the time of fish mortality or early during infection; this may provide useful alternatives for estimating disease progression.

The mechanism of virus release from infected fish populations may be more complex, as farms cause virus entry/infection by multiple routes through contaminated seawater rather than artificial infection. In fact, it has been reported that RSIV was detected in the farm seawater of RSIVD-infected red sea bream, which could spread infection to other fishes located nearby [36]. Although the impact of fish on the disease will be different depending on the current and physical distance of the environmental waters, continuous monitoring in the host animals and seawater will be useful for estimating the risk of transmission between farms in the future by identifying the virus spread model.

Although this study cannot exclude the possibility that the level of viral shedding was lower than the detection limit of the virus flocculation method and real-time PCR, we could not provide direct evidence of virus shedding at 21 and 30 dpi at 15 °C (10^4^ RSIV genome copies/fish). However, we succeeded in detecting RSIV in seawater when the virus was present in very small numbers, that is, 10^1.86^ RSIV genome copies/L seawater, which is a level close to the actual threshold, and we confirmed the change in the concentration of the virus released into the rearing seawater depending on disease progression. These results indicate that viral shedding levels in rearing waters can be indicative of the stage of infection and offer the potential for RSIV monitoring in a non-invasive manner via seawater in aquaculture fields.

## 5. Conclusions

Collectively, the findings of this study show that RSIV shedding into rock bream-rearing seawater varies depending on the viral load, suggesting that RSIV viral shedding could potentially be used for non-invasive detection of RSIV in fish farms.

## Figures and Tables

**Figure 1 animals-12-01978-f001:**
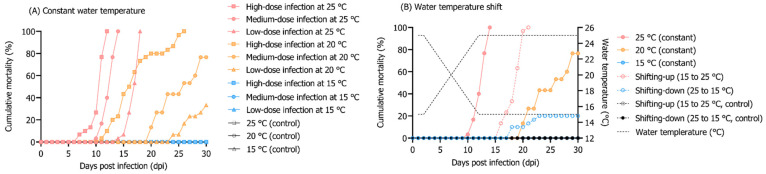
(**A**) Cumulative mortality of rock bream individuals injected intraperitoneally (IP) with red sea bream iridovirus (RSIV) at three concentrations (high-dose infection, 10^8^ RSIV genome copies/fish; medium-dose infection, 10^6^ RSIV genome copies/fish; low-dose infection, 10^4^ RSIV genome copies/fish) at different water temperatures (25 °C, 20 °C, and 15 °C). (**B**) Cumulative mortality depending on the water temperature shift. The water temperature shift group was maintained at 15 °C or 25 °C and IP-injected with 10^6^ RSIV genome copies/fish, and the water temperature was increased or decreased (1 °C/day) three days post infection (dpi). The negative control was IP-injected with 0.1 mL of phosphate-buffered saline (PBS).

**Figure 2 animals-12-01978-f002:**
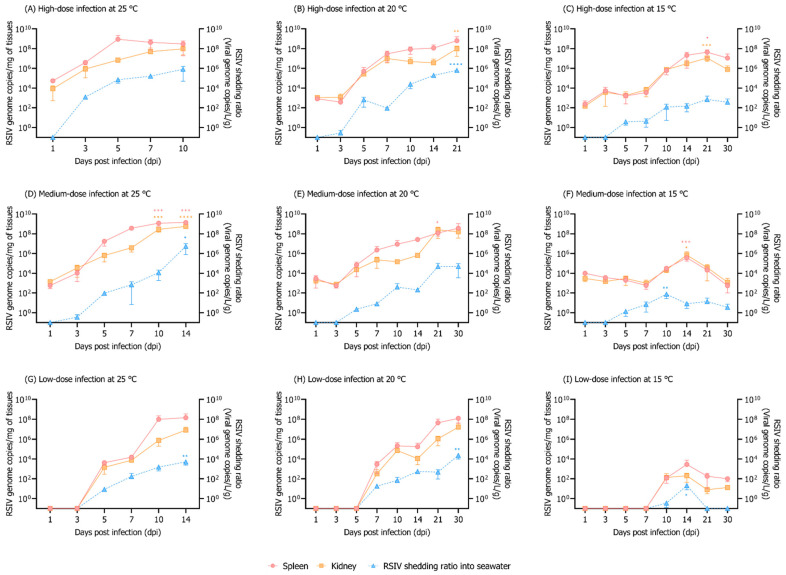
Viral copy numbers in the spleen, kidney, and rearing seawater after virus injection into rock bream at different rearing water temperatures and RSIV concentrations. (**A**–**I**) Rock bream individuals were IP-injected with high- (10^8^ RSIV genome copies/fish), medium- (10^6^ RSIV genome copies/fish), and low-dose infection (10^4^ RSIV genome copies/fish) at 25 °C, 20 °C, and 15 °C. The virus shedding ratio (viral genome copies/L/g) was determined based on the total weight (g) of the fish remaining in the tank and the number of viral genome copies detected in the rearing seawater. The error bars represent the standard deviation of the mean (*n* = 3). Significant differences were determined using a one-way ANOVA with Dunnett’s multiple comparisons test (* *p* < 0.05; ** *p* < 0.01; *** *p* < 0.001; **** *p* < 0.0001).

**Figure 3 animals-12-01978-f003:**
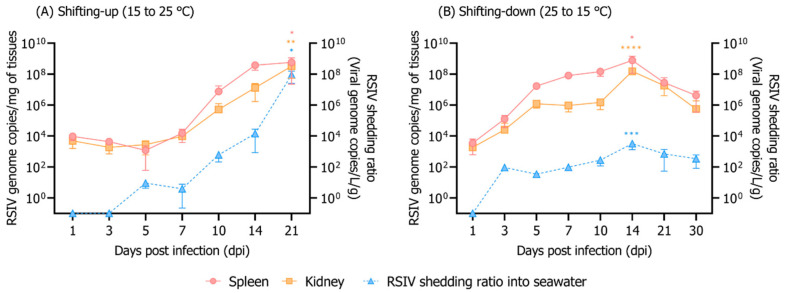
Viral copy numbers in the spleen, kidney, and rearing seawater. Rearing water temperature was adjusted after the rock bream was infected with RSIV injection (10^6^ RSIV genome copies/fish). (**A**) Rearing water temperature of RSIV-infected rock bream at 15 °C, with the temperature increasing to 25 °C starting from 3 dpi (1 °C/day). (**B**) Rearing water temperature of RSIV-infected rock bream at 25 °C, with the temperature decreasing to 15 °C starting from 3 dpi (1 °C/day). The virus shedding ratio (viral genome copies/L/g) was determined based on the total weight (g) of the fish remaining in the tank and the number of viral genome copies detected in the rearing seawater. The error bars represent the standard deviation of the mean (*n* = 3). Significant differences were determined using a one-way ANOVA with Dunnett’s multiple comparisons test (* *p* < 0.05, ** *p* < 0.01, *** *p* < 0.001, **** *p* < 0.0001).

**Figure 4 animals-12-01978-f004:**
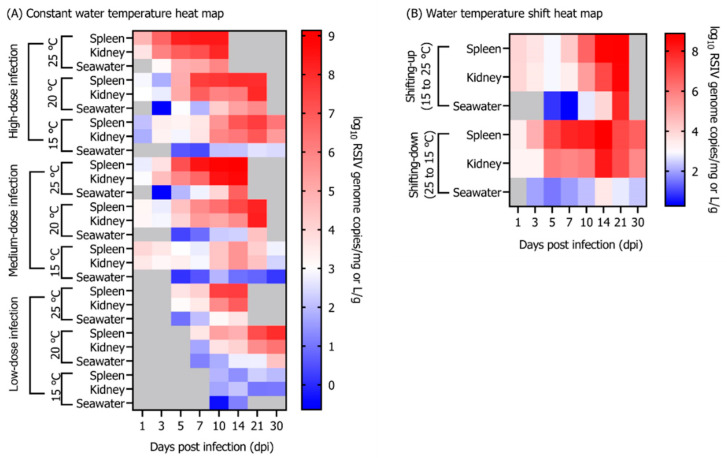
Heat map showing the average virus copy number in the spleen and kidney and the virus shedding ratio into seawater. (**A**) Rock bream individuals were injected IP with high- (10^8^ RSIV genome copies/fish), medium- (10^6^ RSIV genome copies/fish), and low-dose infection (10^4^ RSIV genome copies/fish) at 25, 20, and 15 °C. (**B**) Rock bream was infected with RSIV (10^6^ RSIV genome copies/fish) injection. In the shifting-up group (15 to 25 °C), the rearing water temperature was increased (1 °C/d) from 3 dpi until it reached 25 °C at 12 dpi. In the shifting-down group (25 to 15 °C), the rearing water temperature was decreased (1 °C/d) from 3 dpi until reaching 15 °C at 12 dpi. The virus shedding ratio (viral genome copies/L/g) was determined based on the total weight (g) of the fish remaining in the tank and the number of viral genome copies detected in the rearing seawater. The gray boxes were not analyzed in case of the absence of fish because RSIV was not detected or all fish died.

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
