# Peer review of "Red Sea Bream Iridovirus (RSIV) Kinetics in Rock Bream (Oplegnathus fasciatus) at Various Fish-Rearing Seawater Temperatures"

_animals, 2022, doi:10.3390/ani12151978_

Round 1

Reviewer 1 Report

Dear Authors,

I have performed the review and included my comments in the attached pdf file.

My overall opinion is that the paper needs further improvement. There are a couple of issues that the authors need to address in several sections (e.g. introduction, methods, results, discussion), besides others (e.g. English language and style, quality of figures).

Sincerely yours,

The reviewer

Author Response

We are pleased to verify and rectify problems in the article with the help of your suggestions. We sincerely thank you for your review.

The sentences that require corrections according to your comments have been completely rewritten. We would appreciate it if you could review these corrections.

Responses to some of your comments are as follows:.

  1. Abbreviations for the word “intraperitoneally” (IP) in Materials and Methods subsection 2.2. It was first mentioned when mentioning the virus content.
  2. The country of affiliation of the Bioneer company has been added at the first mention in Materials and Methods subsection 2.3.
  3. In Materials and Methods, TL and GB buffers in subsection 2.4. are the unique names of reagents written in the protocol provided by Bioneer.
  4. The UV device we used (model name: P402) was manufactured by Samji Tech in Korea. It was specially made to order according to the specifications we wanted. Below is the website address with details about these UV devices. http://www.aqt.co.kr/index/bbs/board.php?bo_table=s6_05&wr_id=1
  5. What do you think is the difference in meaning between acclimatized or acclimated? I think it depends on the style of English used in each country. I'm sorry but I would be very grateful if you could explain this.

Thank you very much for taking the time to review our manuscript.

Reviewer 2 Report

The paper is interesting and fluently written. The only problem is the lack of statistical tests and it is not clear how many replicas have been made. I suggest the authors do a PCA and GLMM analysis by using different temperatures, controls and organs. For these reasons I ask the authors to make an effort to bring the paper into scientific standards.

Author Response

We are pleased to verify and rectify problems in the article with the help of your suggestions. We sincerely thank you for your review.

The sentences that require corrections according to your comments have been completely rewritten. We would appreciate it if you could review these corrections.

Responses to some of your comments are as follows:.

We fully agree with your suggestions regarding PCA and GLMM analysis. Samples were obtained and analyzed at different times for each group, and data may not be available for analysis because of fish mortality occurring at different times in each group. In addition, RSIV was not detected at a specific time, and we thought that it is difficult to apply the statistical analysis you suggested because of the lack of data due to early mortality. Thus, we performed a statistical analysis of the change in virus copy numbers with disease progression based on the first detection time point for each experimental group. Thank you very much for your suggestions.

Round 2

Reviewer 1 Report

Dear authors,

Thank you very much for your corrections to the first version of the manuscript and for your replies/corrections.

They have improved the overall output of the manuscript. 

Regarding your question made in reply to one of my comments, i.e. related to "acclimatized versus acclimated":

"Nature/natural environment versus Laboratory conditions"

Once more thank you,

Sincerely yours,

The reviewer

Reviewer 2 Report

No comments !! Now the paper is ok